# The gap between desired and expected fertility among women in Iran: A case study of Tehran city

**Maryam Hosseini, Udoy Saikia[ORCID]\*, Gouranga Dasvarma[ORCID]**

College of Humanities, Arts and Social Sciences, Flinders University, Adelaide, South Australia

\* udoy.saikia@flinders.edu.au

## Abstract

The 2016 Iranian Census reveals that 14 of the country's 31 provinces have sub-replacement fertility. The province of Tehran, where a woman on average gives birth to 1.5 children during her reproductive period, has the lowest fertility in Iran. However, the 'two-child' norm prevails in the country and even a woman of young reproductive age still values having at least two children on average. In other words, there exists a gap between a woman's actual and desired fertility. This paper examines the demographic and socio-economic factors influencing the gap between actual and desired fertility in Tehran city based on a sample survey of 400 married women aged 15–49 years, conducted in 2015. The findings of the study reveal that the women of Tehran would be able to meet their fertility desires of two or more children if they were able to achieve their intended number of children they stated in the survey. However, should these women face any socio-economic constraint, they would be very likely to restrain their fertility desires and have fewer additional children than they initially intended, and thus continue with the sub-replacement fertility as being observed in Iran today.

## Introduction

Stability and change in fertility desires and expectations of women over time have been studied both for possible population forecasting and for their intrinsic merit as indicators of women's attitudes about their future fertility [1]. Factors associated with fertility desires and actual fertility as well as the relationships between them, have been investigated in a number of studies [2–5]. The literature on predictability of reproductive desires, for example [6–9], clearly shows that many individuals do not realise their fertility intentions (desired family size) and therefore, there is often a disconnect between desired and actual fertility.

Understanding the gap between desired family size and actual fertility is also important as it focuses on the reasons behind the inability of women to achieve their stated desired number of children. In addition, it has implications for policy and program interventions related to family planning [2, 7, 10]. According to Bongaarts [11], in societies in early- and mid-demographic transition, desired family size is usually lower than actual family size, whereas the reverse is

Officer at narmon.tulsi@flinders.edu.au) for researchers who meet the criteria for access to confidential data.

**Funding:** The first author received an Australian Postgraduate Research Scholarship to pursue PhD studies at Flinders University under the Flinders University, Project no. 6748 approved on 24 February 2015. This paper is a part of that PhD study.

**Competing interests:** No authors have competing interests.

often the case in post-transitional societies. As argued by Bongaarts [12, 13] and Goldstein et al. [14], fertility in these post-transitional societies would not be below replacement level if couples could realise their fertility intentions.

Iran completed its demographic transition, reaching replacement fertility level of 2.1 children per woman around 2000 [15, 16]. By 2011, 22 provinces had TFRs (total fertility rates) below replacement level [17]. Despite the fact that the findings of the 2016 Census revealed an increase in the total fertility rate for the country from 1.8 children per woman in 2011 to 2.01 children per woman in 2016, 14 provinces still had total fertility rates below replacement level [18]. In other words, these provinces have gone against the Two-Child Norm which is still prevalent in the country [19–22]. A comparison of three birth cohorts of women (born in the 1980s, 1970s and 1960s) in Iran shows that most of these women, including those of the youngest generation still value having more than two children [22]. Therefore, an observed sub-replacement level fertility (less than two children) despite a desire for two children in some provinces of Iran including Tehran province, indicates a disjuncture between desired and actual fertility of Iranian couples.

Studies examining the reproductive behaviour of Iranian couples show that women aged 15–29 and 30–39 years delay having their first and second children [19, 23]. Delaying marriage due to socio-economic constraints on the one hand and infecundity due to delay in childbearing (especially among women aged 35 and older) on the other [9], affect the number of children women have in their reproductive period. As such, young women may end up not realising their desired fertility.

What exists in the whole of Iran can be found on a smaller scale in Tehran city which represents very well the combination of various ethnic and cultural groups of the country [24]. Therefore, in this study the gap between desired and actual family size for Iran will be investigated based on data collected in Tehran city, where women, at the end of their childbearing period generally have fewer children than they initially desired. The total fertility rate in Tehran is 1.56 [18], but couples of Tehran desire to have two or more children [25–28]. It should be noted that economic constraints have strongly been associated with fertility decline in Iran [29]. Therefore, it is not surprising to see such gap in the capital city where the cost of living is significantly higher than other parts of the country [30, 31].

It is within this context that this paper examines the difference between desired and actual number of children in Tehran city, using the data collected in a survey of 400 married women aged 15–49 in 2015. Since a majority of the women surveyed had not completed their reproductive period, the term 'actual fertility' is replaced with 'expected family size' and the term 'unmet fertility desires' is used to refer to the gap between actual and desired fertility of these women. Expected family size is defined on page four of this paper.

## Fertility transition and the emergence of below replacement fertility in Iran

The Islamic Republic of Iran has had one of the most successful family planning programs among developing countries and experienced the fastest fertility decline which has not been widely recorded elsewhere [32, 33]. The first national family planning program in Iran was established in 1967 [34] in order to improve and promote the physical, mental and socio-economic welfare of the family and to reduce the annual population growth rate in the country [35]. Between 1966 and 1976, the total fertility rate in Iran decreased from around 7.7 [36] to around 6.0 [15, 37]. Shortly after the Islamic revolution of 1979, the family planning program was undermined by the new government, and early marriage and large family formation were promoted as basic Islamic values [38, 39]. As a result of these changes and because of the

underlying social, cultural, and political circumstances, fertility increased to around seven children per woman by 1986 [40]. The eight-year (1980–1988) war between Iran and Iraq gave further impetus to pronatalist population policies because a large population was considered advantageous for the nation. The results of the 1986 census revealed that despite the large losses during the war, Iran had attained the highest population growth rate of 3.9 percent per annum ever recorded, comprising 3.2 percent natural increase and 0.7 percent immigration [41]. However, after several campaigns in 1988 aimed at controlling the country's rapid population growth, the government officially revived its voluntary family planning program in 1989 [33]. Thereafter, Iran's total fertility rate (TFR) fell sharply from 5.5 in 1988 to 3.6 in 1993 and then to below 2.8 in 1996 [15, 33]. The Iran Demographic and Health Survey of 2000 (IDHS 2000) showed that the country's TFR dropped further and reached near replacement level (2.26) during 1998–2000 [15, 33]. Iran's fertility rate continued to decline and reached 1.8 by 2006–2011 [42]. However, according to the Iranian Population Census of 2016, the country's total fertility rate appears to have increased slightly to 2.01 [18], but still hovers around replacement level fertility. Further, according to the 2016 Population Census of Iran, 14 of the country's 31 provinces recorded below replacement fertility [18], and the mean TFR in urban areas remained below replacement level at 1.86 births per woman.

It is worth noting that despite the success of the country in lowering its fertility and population growth rate, improving maternal and child health, and in reducing population pressure on its economy, the concerns of politicians and policy makers have now turned to the long-term consequences of below replacement fertility, such as ageing and shortages of working-age population in the future, which has led them to revert to a pronatalist population approach for the country. However, although no such pronatalist policy has yet been officially introduced in the country, funding for population control programs has been drastically curtailed [43] and having a large family is being explicitly promoted in the country. Despite all this, Iran's fertility continues to remain at or around replacement level.

## Research setting

In Iran, there is a persistence of low fertility in a pronatalist environment. Women or married couples may desire larger families but do not achieve what they desire. The concept of desired family size and its importance for understanding reproductive behaviour has been examined in various studies. In fact, changes in fertility desires still play an essential role in most theories of fertility decline [10, 44–46]. Desired fertility or desired family size should not to be mixed up with ideal family size, although in some studies the two are considered one and the same [6, 7, 22, 47]. According to the Multilingual Demographic Dictionary [48 p80] a distinction is made between *desired family size*, which is the number of children a woman, man or couple wants to have, and *ideal family size* which a woman, man or couple envision for their society. The present research is concerned with desired fertility or desired family size as defined above.

Easterlin defines desired family size as 'the number of children parents would have if there were no subjective or economic problems involved in regulating fertility [49].

Goldstein et al. [14] state that in the previous rounds of the Eurobarometer Survey (1979 and 1989) questions about family size were ambiguous and it was not clear whether they were asking the respondents' own family size or that of the respondents 'larger society' [14]. Testa [50] considers that the definition of personal ideal family size is consistent with that of desired family size.

There are two ways to ask about desired fertility in cross-sectional studies. One way is to ask about the number of children a woman says that she would have by the end of her childbearing period if she could start her married life all over again [44]. The other way of querying

a woman about her desired fertility could be to ask about her desired number of children at the current stage of her reproductive span. Both these methods were employed at the Eurobarometer Surveys of 2001, 2006 and 2011 mentioned above.

Consequently in this study, following Easterlin [49]; the surveyed women were asked about their current desired number of children with the following question:

"For you personally, what would be the number of children you would like to have in the absence of any possible obstacle?"

A problem that could arise from asking desired family size in the past is that respondents (especially those aged 40–49) may have forgotten what they had desired earlier in their married life, or they may state their current number of children as their desired family size. The last mentioned possibility would give rise to so-called *post-facto* rationalisation. This is true to some extent in the present study where women with larger number of children ever born or still alive tended to state a slightly higher desired fertility in the future (although this could also reflect the norm for these cohorts of women), but the predominant desire for desired future fertility tended to be centralized on two children. Calculations based on survey data (not reproduced here) show that overall 63% of all the women stated two or fewer children as their desired fertility, with 67% and 40% of the women with two or fewer living children and more than two living children respectively stating two or fewer children as their desired fertility. However, this slight tendency of *post-facto* rationalisation is also found among the women when they were asked about how many more children, they intended to have given their current and possible future socio-economic situations. As will be explained later, additionally intended number of children forms the basis of calculating women's "expected" fertility, a measure that is used to replace women's actual fertility. It is contended that the so-called *post-facto* rationalisation in the two measures (desired fertility and expected fertility) would tend to cancel out each other when examining the gap between the two.

It is worth noting that respondents in this study are aged 15–49 years with a median age of 34. Therefore, a majority still have years to complete their reproductive span. Despite the fact that investigating the gap between actual (observed) fertility and desired fertility may be more realistic, in this study the term 'actual fertility' is replaced with 'expected family size' because of a yet incomplete fertility of young respondents. Consistent with Goldstein et al. [14], expected family size is defined in this paper as:

*'The sum of the number of children a woman has already had at the time of the survey and the number of children the respondent still plan[s] to have'.*

To achieve the number of children the respondent plans to have, which is called 'the additionally intended family size' [50], respondents were asked the question:

*'In addition of the number of children you already have, how many (more) children you have intended to have in the rest of your childbearing period?'*

To answer this question, the respondents needed to think more correctly about their own situation. They should take into account the obstacles that might interfere with achieving desired family size. Considering the difficulties women may face in achieving their desired family size, according to Goldstein et al. [14], the expected family size is considered to be universally smaller than the desired number of children. The gap between desired and expected

family size is calculated by subtracting the woman's desired number of children from the expected number of children.

Following Quesnel-Vallée and Morgan [9], the expected-desired fertility gap is categorised in this paper into three groups as follows:

i. Underachieved. When the desired family size is more than the expected family size. In such a situation the expected-desired fertility gap would be negative (gap<0) and would indicate that women have unmet fertility desires.

ii. Achieved. When the desired family size equals the expected family size. In this case, the expected-desired fertility gap would be zero (gap = 0), indicating that women's fertility desires have been met.

iii. Overachieved. When desired family size is less than expected family size, resulting in a positive gap between expected and desired fertility, indicating that in case women have overshot their fertility desires.

Various factors are expected to influence the gap between desired and expected family size. However, in the present study, variables such as women's age, age at marriage, sex composition of children, consanguineous marriages between two cousins (cousin marriage), educational attainment, employment status and contraceptive use are selected as independent variables influencing the fertility gap.

## Research methods

This research was conducted with due observance of research ethics involving human subjects. It was approved in Australia by the Social and Behavioural Research Ethics Committee, Flinders University, Australia. Project no. 6748 approved on 24 February 2015. Further, as per ethics committee guidelines the participants in this research were assured of their anonymity and the confidentiality of the information they gave, and written consent was obtained from each participant. This paper is extracted from the PhD dissertation of the first author written under the supervision of the co-authors at Flinders University, Australia. It is predominantly based on primary data collected through a fieldwork survey in Tehran city (the capital of Iran). It is worth noting that the population of Tehran city by 2016 has reached 8.6 million which represents 14.7 percent of the urban population of the whole country. Tehran has one of the lowest TFRs in Iran. The total fertility rate in capital city which was stable around 1.3 children per woman for one decade (1996–2006 and 2006–2011), has slightly increased and reached 1.5 children per woman in the latest census (2016). Women of Tehran have the highest mean age at marriage in the country [27]. Moreover, in Tehran city a significant proportion of women aged 20–49 has at least university degrees [18].

According to the 2011 National Census of Iran (the latest census before conducting this survey) the total population of Tehran city was approximately 8.15 million, which consists of 1.57 million married women of childbearing ages 15–49 years. This population lived in 2.83 million households [42]. The present study is based on information collected from a sample of the population of these women. Due to time and resource constraints, it was not possible to select a larger sample which could be fully representative of the study region. However, all attempts have been made to minimise any bias in the sample selection. It should be noted that in this study only married women were interviewed because as in many other Muslim countries, premarital cohabitation and out of wedlock childbearing is prohibited in Iran [51, 52] so that almost all births may be considered to occur within marriage. The sample size was determined by using Cochran's formula [53]. With these specifications, the sample size was calculated at

384. However, a round figure of 400 was taken to account for possible non-responses. According to the 2011 National Census of Iran, the average household size of Tehran province is 3.3 [42], therefore it was assumed that on average there would be at least one married woman aged 15–49 years in a household. Thus, households were selected first and from each selected household, one currently married woman of reproductive ages 15–49 years was chosen at random as an eligible respondent. If a selected household had no eligible woman living there, the interviewer chose the next household on the list of sampled households which did have an eligible woman.

Thus, in this study around 400 ever-married women aged 15–49 years were interviewed with the help of structured questionnaires developed for this purpose. As mentioned earlier, these women represent the various ethnic and cultural groups of Iran.

The overall theoretical framework used in this paper is derived from Quesnel-Vallée and Morgan [9] built on Bongaarts' (2001; 2002) model explaining the inconsistency between desired and actual fertility behaviour among women.

Although in Bongaarts' model [12, 13] fecundity impairment and unwanted fertility can result in inconsistencies between desired and actual fertility, they are not used as explanatory variables in this study, because information on infecundity and unwanted fertility is not available. Therefore the variables eventually selected as independent variables are age, age at first marriage, sex composition of current (surviving) children, consanguineous marriage, level of education, employment status and contraceptive use.

A bivariate analysis showing the relationship of the gap between expected and desired family size with explanatory variables taken one at a time is investigated in terms of Chi-Square values and their statistical significance. However, since all the independent variables selected for analysis could simultaneously influence the fertility gap, a multivariate analysis of the influence of all the independent variables taken together on the dependent variable is deemed necessary. In this study, married women aged 15–19 years are excluded from the analysis as their proportion is very small (only one percent). The results and discussions presented below exclude the variables which did not exhibit a statistically significant relationship with the gap between expected and desired family size.

## Empirical findings from the bivariate analysis

### The relationship between age and the expected-desired fertility gap

The fertility behaviour and intention of women as well as the estimated gap between women's expected and desired family size (in terms of underachieved, achieved and overachieved) based on different age groups of women are shown in Table 1. As expected, the mean number of children ever born to women shows a rising trend with age, with values increasing from 0.74 to 2.16 children among surveyed women aged 20–29 to 40–49 respectively. This is obvious because older women are exposed to the risk of conception and childbearing for a longer period than younger women, therefore they are expected to give birth to more children than younger women.

Because of the dynamic nature of the children ever born (CEB) and additionally intended numbers of children, the expected family size varies by women's age. As mentioned earlier, expected family size is the sum of the actual (CEB) and the additionally intended number of children. According to Testa [50], at the beginning of the reproductive span the expected family size reflects mainly the additionally intended number of children. However, at the end of women's reproductive span expected family size reflects mainly the actual family size [50 p8]. The findings of this paper show that the expected family size, increases with women's age (Table 1). Women aged 40–49 expect to have more than two children (2.38) as their lifetime

**Table 1. Mean number of children ever born (CEB), mean expected CEB, mean desired family size and the gap between mean CEB and mean expected CEB by current age.**

| Age group | Number of women | Mean CEB | Mean additionally intended number of children | Mean expected CEB | Mean desired family size | Gap | | | $\chi^2$ and p-value | Cramer's V |
|---|---|---|---|---|---|---|---|---|---|---|
| | | | | | | Negative (Underachieved desired fertility) | Zero (Achieved desired fertility) | Positive (Overachieved desired fertility) | | |
| 20–29 | 102 | 0.74 | 1.01 | 1.85 | 2.34 | 35.5 | 59.8 | 4.7 | 17.45 | |
| 30–39 | 181 | 1.29 | 0.6 | 1.99 | 2.36 | 41.4 | 49.2 | 9.4 | | 0.14 |
| 40–49 | 113 | 2.16 | 0.14 | 2.38 | 2.62 | 37.5 | 42.0 | 20.5 | P < .005 | |

Ever married women aged 20–49 years. Tehran city. 2015.

*Note*: The gap has been calculated at individual levels. It means the gap is calculated by subtracting the desired CEB from the expected CEB. Therefore, mean gap = mean (Desired CEB-expected CEB).

*Source*: [54].

fertility, however the expected family size among women aged 20–29 and 30–39 are 1.85 and 1.99 per married women, respectively. As shown in Table 1 there is a positive relationship between women's age and their desired family size, although the association is weak, as indicated by the small values of $\chi^2$ and Cramer's V. Considering the fact that the three age groups selected in this study represent three different birth cohorts, the figures of Table 1 reflect changes in desired fertility by generation. As Goldstein et al. [14] discuss, the younger cohorts are more likely to prefer smaller family sizes than older ones. In fact, the older cohorts grew up in a different time and were surveyed at an older age. It may be that the desires of the younger cohorts, because of life-cycle influences, will approach those of the older cohorts as they age [14].

The findings of this paper also show that the highest mean desired family size is seen among women aged 40–49, which is 2.62 children per woman (Table 1). Interestingly, the desired family size among women aged 20–29 and 30–39 is almost the same (2.3). The marriage and the start of childbearing of these two age cohorts have been coincident with the stability of the total fertility rate below the replacement level (1.8) for almost one decade in the country (from 2000–2006 and 2006–2011) [42, 55]. Despite the fact these young cohorts grew up and got married when the fertility had dropped to below replacement level in Iran, the desired family size among women aged 20–29 and 30–39 is above two children. This is consistent with Bongaarts [11] who argues that in post transitional societies the desired family size exceeds the fertility outcomes. Results shown in Table 1 also demonstrates the gap between expected and desired family size in terms of underachieved, achieved and overachieved. As can be seen in in the table, there is a significant relationship between women's age and the gap between their expected and desired family size (Chi square with p < .005). A test of the adjusted residuals (not shown here) indicates that the significant association between women's age and the gap is mostly driven by the cell of overachieved category. Further, as can be seen in Table 1, a large proportion of women in each age group achieve their desired fertility. However, it should be noted that one of the components of the gap is the expected family size which depends on the women's future childbearing plans (additionally intended number of children). In case of any lifetime obstacle i.e. primary or secondary infertility or marriage disruption, younger age groups may not be able to reach the additional number of children they have already attained. Again, as mentioned earlier, because of the dynamic nature of the number of children ever born and the additionally intended number of children with respect to age of the women, the fertility gap among the younger women might change as these women grow

older, but the gap among the oldest women (aged 40–49 in this case), who have already completed their fertility is not likely to change.

## Relationship between age at first marriage and the expected-desired fertility gap

Age at marriage is one of the primary proximate determinants of fertility which directly impacts on women's fertility behaviour [56]. It has been argued that delayed marriage reduces fertility compared to what is initially desired [9, 57]. Consistently, the results of the Chi-Square test in this paper show that there is a very significant negative relationship between women's age at first marriage and the number of children ever born to women (p<0.005) (Table 2). It means the older the women are at the time of their marriage, the fewer would be the number of children ever born to them. Women who married at younger ages are exposed to a sexual relationship for a longer period. Therefore, they are likely to have a higher fertility than women who get married at older ages. Moreover, the age-related declines in fecundity (especially after age 35) [9] should be taken into account as a factor reducing the fertility of women who have delayed marriage. The strong relationship between women's age at first marriage and fertility is more evident in countries in which the extra-marital childbearing is culturally or religiously prohibited [58]. According to a study by Abbasi-Shavazi et al. [16], the increase in age at marriage in Iran, which is a result of increases in women's education, has been one of the significant factors affecting the fertility decline in the late 1980s in the country. As mentioned earlier, the expected number of children is highly related to the length of women's reproductive period, therefore, follows the same pattern as CEB among women with different ages at marriage.

The findings of this study, using Chi-Square test reveal that there is a positive and statistically significant relationship between women's age at first marriage and the expected-desired fertility gap (Table 2). Interestingly, as can be seen in the table, a larger proportion of women in each category of age at marriage have achieved their desired fertility. Table 2 shows that the proportion of women with overachieved fertility in the oldest category of age at marriage (older than 25), is significantly less than the other categories (younger than 19 and 20–24). Women who have had delayed marriages have significantly higher levels of education than women in other marriage age-groups. Moreover, a majority (60 percent) of these women are employed and have a professional career, therefore it can be concluded that these women have

**Table 2. Mean number of children ever born (CEB), mean expected CEB, mean desired family size and the gap between mean CEB and mean expected CEB by age at first marriage.**

| Age group (age at first marriage) | Number of women | Mean CEB | Mean additionally intended number of children | Mean expected CEB | Mean desired family size | Gap | | | $\chi^2$ and p-value | Cramer's V |
|---|---|---|---|---|---|---|---|---|---|---|
| | | | | | | Negative (Underachieved desired fertility) | Zero (Achieved desired fertility) | Positive (Overachieved desired fertility) | | |
| <19 | 112 | 1.93 | 0.31 | 2.34 | 2.67 | 39.7 | 42.1 | 18.2 | 82.05 | 0.32 |
| 19–24 | 139 | 1.47 | 0.56 | 2.14 | 2.42 | 36.7 | 51.8 | 11.5 | P<0.05 | |
| >= 25 | 145 | 0.82 | 0.88 | 1.76 | 2.25 | 42 | 53.1 | 4.9 | | |

Ever married women aged 20–49 years. Tehran city, 2015.

Note: The gap has been calculated at individual levels. It means the gap is calculated by subtracting the desired CEB from the expected CEB. Therefore, mean gap = mean (Desired CEB-expected CEB).

Source: [54].

more autonomy to restrict their fertility and have more knowledge of contraceptive use to prevent unwanted fertility and consequently prevent overachieved fertility.

## Sex composition of children and the fertility gap

It is well documented that there exist preferences for particular sex of their children among parents in many countries, particularly in developing countries [59–62]. Some studies argue that sex preference has a marginal influence on fertility; however, others claim that sex preference has augmented fertility rates in the past and would provide a barrier to future fertility declines [63].

A preference for sons has been reported in many Eastern and Southern Asian countries and such preference results from traditional religious beliefs, economic benefits or social reasons [62, 64, 65]. The degree of son preference varies substantially from one country to another depending on the level of economic development, social norms, cultural and religious practices, marriage and family systems, degree of urbanization, and the nature of social security systems [66]. The influence of sex preference on fertility behaviour in Iran has been investigated in several studies. For example, Abbasi-Shavazi, McDonald [67] argue that sex preference in Iran, even in cities with more traditional values is disappearing gradually. They show that increases in urbanization and structural changes in agriculture have led to reduced need for manual labour and therefore there is less desire for having sons to strengthen the workforce. However, Mahmoudian and Mahmoudiani [68] argue that although the desire for equal number of boys and girls is common among women and men, the desire for more sons than that for daughters is much prevalent in Iran. Studies in Tehran city, the capital of Iran show that in the context of low fertility, where a majority of women have higher education [69], gender preference is not a driving force in fertility decision-making [28, 70]. Moreover, because of high proportions of educated women and men and the general prevalence of nuclear families in Tehran city, the influence of in-laws or other family members in fertility decision-making is minimal.

In the present study women were also asked about their preference for the sex of their next child (if they have any childbearing plan in the future). It may be noted that women with no live birth (CEB = 0) have been excluded from the table, because there is no sex composition among women of zero parity. It is also worth noting that women of zero parity comprise almost 25 percent (104 women) of the study sample.

The findings of this study show that most of the women who have more girls than boys among their surviving children wish to have a son as their next child and most of the women who have more boys than girls among their surviving children wish to have a girl as their next child. But almost two-thirds of the women who have equal numbers of boys and girls among their surviving children are non-committal about their preference for the sex of the next child.

The fertility behaviour and intention of women based on the sex composition of current (surviving) children is demonstrated in Table 3. As before, women with no live birth (CEB = 0) have been excluded from this table because there is no sex composition among women of zero parity.

The Chi-Square ($\chi^2$) test in Table 3 shows a moderately strong and statistically significant association between the sex composition of surviving children and the 'expected-desired' fertility gap, indicating that a majority of women with balanced sex-composition of surviving children would achieve their desired family size. This finding is consistent with above mentioned findings which shows that there is an intention to have a balanced sex composition of children among women.

The proportion of women who would achieve their fertility desires (gap = 0) is almost the same among women who have an unbalanced sex-composition among their children.

**Table 3. Mean number of children ever born (CEB), mean expected CEB, mean desired family size and the gap between mean CEB and mean expected CEB by sex composition of surviving children.**

| Sex composition of current (surviving) children | Number of women | Mean CEB | Mean additionally intended number of children | Mean expected CEB | Mean desired family size | Gap | | | $\chi^2$ and p-value | Cramer's V |
|---|---|---|---|---|---|---|---|---|---|---|
| | | | | | | Negative (Underachieved desired fertility) | Zero (Achieved desired fertility) | Positive (Overachieved desired fertility) | | |
| Equal number of boys and girls | 63 | 2.06 | 0.11 | 2.17 | 2.73 | 43.4 | 54.8 | 1.8 | 28.53 | 0.19 |
| More boys than boys | 107 | 1.05 | 0.39 | 2.13 | 2.5 | 39.6 | 46.8 | 13.5 | | |
| More girls than girls | 131 | 1.91 | 0.41 | 2.37 | 2.5 | 33.9 | 44.9 | 21.3 | P<0.005 | |

Ever married women aged 20–49 years. Tehran city, 2015.

Note: The gap has been calculated at individual levels. It means the gap is calculated by subtracting the desired CEB from the expected CEB. Therefore, mean gap = mean (Desired CEB-expected CEB).

Source: [54].

However, the proportion of overachievement among women who have more girls than boys is much higher than those who have either a balanced number of children or who have more boys than girls. This finding shows that couples, who have more girls than boys may continue their childbearing to get another male child and end up producing more children than they initially desire.

This may raise the question whether it is important for couples to have or not to have a son in deciding about their desired family size. This question is addressed in Table 4, which shows that there is no statistically significant relationship between having or not having a son and the 'expected-desired' fertility gap. This finding from Table 4 should be interpreted in conjunction with the fact that the association between sex-composition and the 'expected-desired' fertility gap shown in Table 3 is only moderate, as indicated by the values of $\chi^2$ (less than 30) and Cramer's V (less than 0.3). Therefore, as mentioned earlier, it can be concluded that son preference in Tehran city is not a major driving force in fertility decision-making among couples.

The effect of age on women's sex composition of children in this study shows that there is almost no difference in the median age of women with different sex composition of children. It can be concluded that, in this study the age factor has not affected the sex composition of surviving children of women.

**Table 4. Mean number of children ever born (CEB), mean expected CEB, mean desired family size and the gap between mean CEB and mean expected CEB by sex composition of children based on having or not having a son.**

| Sex composition of children based on son | Number of women | Mean CEB | Mean additionally intended number of children | Mean expected CEB | Mean desired family size | Gap | | | $\chi^2$ and p-value |
|---|---|---|---|---|---|---|---|---|---|
| | | | | | | Negative (Underachieved desired fertility) | Zero (Achieved desired fertility) | Positive (Overachieved desired fertility) | |
| Not having a son | 99 | 1.47 | 0.44 | 2.03 | 2.48 | 41.4 | 49.5 | 9.1 | 3.95 |
| Having at least one son | 202 | 2.04 | 0.29 | 2.35 | 2.58 | 34.8 | 47.8 | 17.4 | p>0.05 |

Ever married women aged 20–49 years. Tehran city, 2015.

Note: The gap has been calculated at individual levels. It means the gap is calculated by subtracting the desired CEB from the expected CEB. Therefore, mean gap = mean (Desired CEB-expected CEB).

Source: [54].

## The relationship between women's education and expected-desired fertility gap

Over the past few decades, many developing countries have adopted policies designed to reduce rapid population growth. Educating young women is one of these policies and is considered highly effective in achieving this goal [71]. The inverse relationship between the level of education and fertility is shown in a number of studies [72–80]. Several explanations for the influences of female education on fertility have been provided by economic theories of fertility [81]. The increased opportunity costs of childbearing and child rearing among educated women was a topic discussed by Becker [81] and Schultz [82]. Moreover, it has been considered that education may lower fertility levels through improvements in child health and reduced rates of child mortality because women need to have fewer births to meet the same desired number of children [83, 84]. Female schooling also can affect fertility through increasing female autonomy and empowerment in fertility decision making [85]. Improvement in women's knowledge about effective methods of contraception is another impact of female education on fertility [86, 87].

Even though education mostly has had an attenuating impact on fertility, in some cases it has the potential to raise it. According to Lesthaeghe et al. [88] modest improvements in female education in some less developed countries were shown to increase fertility slightly, however, even at low levels of socio-economic development, a negative association emerged after a critical level of schooling.

The findings of the present study in Table 5 reveals that in line with some previous studies, there is a negative and statistically significant relationship between women's level of education and CEB. Further, it is also noted from data collected in this study that the level of women's education is negatively associated with their current age. In other words, younger women tended to attain higher levels of education compared to older women. This helps understand the findings of Table 1 which shows that younger women, (and therefore more educated women) have the lower CEB and lower expected family size compared with older women with lower levels of education.

Some cross-national studies show that the impact of women's education is much stronger than men's in reducing fertility [89, 90]. In fact, educated women face greater opportunity

**Table 5. Mean number of children ever born (CEB), mean expected CEB, mean desired family size and the gap between mean CEB and mean expected CEB by women's level of education.**

| Level of education | Number of women | Mean CEB | Mean additionally intended number of children | Mean expected CEB | Mean desired family size | Gap | | | $\chi^2$ and P value | Cramer's V |
|---|---|---|---|---|---|---|---|---|---|---|
| | | | | | | Negative (Underachieved desired fertility) | Zero (Achieved desired fertility) | Positive (Overachieved desired fertility) | | |
| Elementary school | 88 | 1.98 | 0.38 | 2.4 | 2.55 | 35.2 | 44.3 | 20.5 | | |
| High school and diploma | 123 | 1.54 | 0.52 | 2.18 | 2.53 | 37.4 | 53.4 | 9.2 | 10.41 | 0.11 |
| Bachelor's degree and above | 185 | 0.97 | 0.75 | 1.8 | 2.29 | 42.5 | 49.2 | 8.3 | P<0.05 | |

Ever married women aged 20–49 years. Tehran city, 2015.

Note: The gap has been calculated at individual levels. It means the gap is calculated by subtracting the desired CEB from the expected CEB. Therefore, mean gap = mean (Desired CEB-expected CEB).

Source: [54].

costs, therefore revise their childbearing downward compared with their earlier intentions. The gender roles and the lack of biological constraints may make the opportunity costs competition less intense among men than women [9].

It is worth noting that in this paper almost 70 percent of women with primary and secondary education are aged 40–49 years and most women aged 20–29 and 30–39 years have tertiary education. Therefore, the findings of Table 5 are consistent with the results of Table 1 which show that the age groups 20–29 and 30–39 share the same desired family size, although women aged 40–49 desire more children. The age pattern of women based on their level of education confirms the fact that the desire to have more children declines with increasing level of education. Therefore, the expected family size which is the sum of the CEB and the additional (similar with the CEB) has a declining trend with increase in the level of education. Coming back to Table 5, using the results of Chi-Square test, indicates that women's educational attainment has a significant relationship with the gap between desired and expected number of children. It is evident from the table, except for the difference in the proportion of women with overachieved fertility, a convergence can be seen among women with different levels of education in terms of achieving their fertility desires. In fact, women with achieved fertility are dominant in each level of education. The lower proportion of women with overachieved fertility among women with the highest level of education is not surprising. Having more autonomy and knowledge over contraceptive use, to restrict fertility and prevent unwanted births, resulted in having more planned pregnancies among highly educated women. However, the interference of the higher opportunity costs of childbearing on the one hand and the age-related decline in fecundity (due to higher age at marriage) on the other, has caused a bigger proportion of educated women to have unmet fertility desires (underachieved fertility) (Table 5).

Conversely, the higher proportion of overachievement among women with primary education, in addition of their lower age at marriage and their lower opportunity costs of childbearing, can be attributed also to their sex composition of children. More interestingly, a significant proportion (44 percent) of women with the least educational attainments, have more girls than boys. Consistent with the discussion in the section of 'sex composition of children' (Table 3) although having a balanced number of male and female children is desired by a majority of women, couples who have more girls (than boys) are more likely to continue their childbearing to get a male child. Therefore, at the end of their reproductive period these women may experience overachieved fertility related to their desires.

## Employment status and its relationship with the expected-desired gap

Women's labour force participation has been investigated in most explanations of fertility. The depressing effect of women's employment on fertility is consistent with numerous studies which, at the individual level, have shown that there is a negative association between fertility and women's labour force participation [91–94]. In fact, both industrialised and developing countries have formulated policies based on the negative association between these two aspects of women's lives [91].

The economic theory of fertility [81] hypothesizes that, within-country, the negative relationship between work and fertility (similar to education and fertility) is a result of the opportunity costs of childbearing and child-raising particularly for highly-skilled women. The rise of women's educational and labour market attachment can result in a fall in women's propensity to have children [81]. This negative correlation may also be due to increasing the financial rewards flowing from postponing parenthood [95, 96].

Consistent with studies (e.g. [97–99]) showing a negative association between women's employment and fertility, the findings of this paper in the following table demonstrates an

**Table 6. Mean number of children ever born (CEB), mean expected CEB, mean desired family size and the gap between mean CEB and mean expected CEB by women's employment status.**

| Employment status | Number of women | Mean CEB | Mean additionally intended number of children | Mean expected CEB | Mean desired family size | Gap | | | $\chi^2$ and p-value | Cramer's V |
|---|---|---|---|---|---|---|---|---|---|---|
| | | | | | | Negative (Underachieved desired fertility) | Zero (Achieved desired fertility) | Positive (Overachieved desired fertility) | | |
| Employed | 179 | 1.09 | 0.66 | 1.82 | 2.26 | 40.1 | 54.4 | 5.5 | 11.18 | 0.17 |
| Not Unemployed | 217 | 1.62 | 0.54 | 2.27 | 2.58 | 38.2 | 45.9 | 15.9 | P<0.005 | |

Ever married women aged 20–49 years. Tehran city, 2015.

*Note*: The gap has been calculated at individual levels. It means the gap is calculated by subtracting the desired CEB from the expected CEB. Therefore, mean gap = mean (Desired CEB-expected CEB).

Source: [54].

inverse relationship between women's employment status and their fertility behaviour and preferences in Tehran city (Table 6). As can be seen the mean CEB is higher among unemployed women than the employed women. Despite the fact the additionally intended number of children is almost similar among both employed and unemployed women, the expected family size among them follows the same trend as the CEB. Therefore, the expected number of children, which is women's lifetime fertility goals, among unemployed women is higher than the expected family size among employed respondents.

Women's educational attainment in Iran have not been translated into the increase in their labour force participation [38]. The findings of the National Population and Housing censuses in Iran, particularly Census 2011, show that women's employment and work force participation has always had a significant negative relationship with fertility in Iran. According to the findings of the 2011 census, women comprised 49.6 percent of population of 10 years old and above, but their labour force participation was only 11.4 percent (Statistical Centre of Iran, 2011).

However, women's economic participation does not necessarily lead to fertility decline in Iran. In fact, having a secured fulltime job along with a high income not only does not lead to fertility decline, but also can result in an increase in women's desire to have more children as their job security and satisfaction give them more confidence to raise a child [100].

The findings of this paper in Table 6 reveal that there is a statistically significant relationship between women's employment status and the fertility gap in terms of underachievement, achievement and overachievement of fertility desires, but this association is weak as indicated by the small values of $\chi^2$ and Cramer's V. However, it may be seen apparently that employed women, notwithstanding their higher opportunity costs of childbearing and accumulated human capital, exhibit a level of prevalence of underachievement of fertility that is no different to that among unemployed women (Table 6). This table also reveals that although almost a large proportion of employed and unemployed women (54.4 and 45.9 percent respectively) would have achieved their fertility desire, the prevalence of overachieved fertility among unemployed women is almost three times than that among employed women.

Interestingly, a Chi-Square test (not shown here) did not show any significant relationship between women's age and employment. Therefore, considering the weak association between age and fertility gap shown in Table 1, it may be said that the age factor cannot be considered to explain the higher levels of overachievement among unemployed women. The correlation between the sex composition of children (as a factor which was highly associated with the

Table 7. The gap between desired number of children and expected number of children by women's parity.

| Women's parity | Gap | | | $\chi^2$ and p-value | Cramer's V |
|---|---|---|---|---|---|
| | Negative (Underachieved desired fertility) | Zero (Achieved desired fertility) | Positive (Overachieved desired fertility) | | |
| 1 | 51.3 | 44.4 | 4.3 | 66 | 0.33 |
| 2 | 33.1 | 56.7 | 10.2 | | |
| 3+ | 16.1 | 37.5 | 46.4 | P<0.05 | |

Ever married women aged 20–49 years. Tehran city, 2015.

Source: [54].

fertility gap), and women's employment status shows that most employed women in this study have a balanced sex composition of their children (almost 50 percent). However, among unemployed women the current sex composition of children is in favour of girls which may result in overachieved fertility, in their desire to have a male child.

Another perspective on fertility gap is to examine the same according to women's parity (Table 7), which shows that the highest proportion of achieving desired fertility appears to be among women of Parity 2. This is in line with previous studies [21, 24] which show that the two-child norm is still prevalent in Tehran city. The highest prevalence of underachieved fertility is found among women of Parity 1, but these women may be younger who have not yet completed their childbearing. The prevalence of overachievement of fertility is found among women of Parity 3 and above. This table might show a trend from underachievement, achievement to overachievement with increasing parity. The association between parity and level of achievement of desired fertility appears to be strong as indicated by the values of $\chi^2$ and Cramer's V.

Among other factors considered pertinent in this study (not shown here), it is found that consanguineous marriages and contraceptive use have no statistically significant association with the gap between desired and actual fertility.

Moreover, it is worth noting that none of the surveyed women in this study reported that cost or distance was a factor in their lack of access to family planning services; rather any non-use of contraception is found to be mainly associated with the intention to have a child, and/or secondary and primary infertility.

## Multivariate analysis of the fertility gap: Findings from the logistic regression model

The bivariate analysis of the relationship between expected-desired fertility gap and the explanatory variables such as age, age at marriage, cousin marriage, sex composition of current (surviving children), educational attainment and employment status revealed that all mentioned variables (except contraceptive use) have a statistically significant association with fertility gap, even though in many cases the strength of the association, as measured by $\chi^2$, is not very strong. Considering the fact that the dependent variable in this study (fertility gap) has three categories -underachieved, achieved and overachieved, therefore, the multinomial logistic regression was conducted to analyse the impact of the explanatory variables taken together, on the likelihood of achieving the fertility goals (Table 8). Since, as shown in the bivariate tables, the significant associations between each of the explanatory variables and the dependent variable (the gap between expected and desired fertility) was mostly driven by the cell 'overachieved' fertility gap, the category 'achieved fertility' of the dependent variable was set as the reference. The findings of the multinomial logistic regression in Table 8 show that all the

**Table 8. Odds of under-achieving or over-achieving fertility with reference to achieved fertility according to demographic and socio-economic variables.**

| Demographic and socio-economic variables | Underachieved fertility | | Overachieved fertility | |
|---|---|---|---|---|
| | **Odds ratio** | **p-value** | **Odds ratio** | **p-value** |
| **Age -group (in years)** | | | | |
| 20–29 | 0.66 | 0.19 | 0.16 | 0.002 |
| 30–39 | 1.02 | 0.94 | 0.49 | 0.09 |
| 40–49 (reference category) | | | | |
| **Age at first marriage (in years)** | | | | |
| < 19 | 1.39 | 0.28 | 4.49 | 0.009 |
| 20–24 | 0.96 | 0.9 | 2.58 | 0.09 |
| > = 25 (Reference category) | | | | |
| **Marriage among relations (Cousin marriage)** | | | | |
| Relative | 0.84 | 0.5 | 1.77 | 0.13 |
| Non-relative (Reference category) | | | | |
| **Sex composition of children** | | | | |
| Equal number of boys and girls | 1.07 | 0.33 | 0.1 | 0.001 |
| More boys than girls | 1.12 | 0.55 | 0.7 | 0.59 |
| More girls than boys (Reference category) | | | | |
| **Level of Education** | | | | |
| Elementary school | 0.64 | 0.19 | 0.64 | 0.44 |
| High school and diploma | 0.66 | 0.14 | 0.37 | 0.04 |
| Bachelor's degree and above (Reference category) | | | | |
| **Employment status** | | | | |
| Employed | 0.83 | 0.45 | 0.4 | 0.02 |
| Unemployed (reference category) | | | | |

Ever married women aged 20–49. Tehran city, 2015.

Source: [54].

explanatory variables show statistically significant effects on the likelihood of women over-achieving their fertility desires (in comparison to achieved fertility).

In terms of the explanatory variables, it is found that women aged 20–29 are 84 percent less likely than women aged 40–49 to have an overachieved fertility in comparison to achieved fertility (Table 8). However, it seems after controlling for other demographic and socio-economic variable the impact of the middle age group (30–39) on the likelihood of overachievement is statistically not insignificant.

The lower likelihood of overachievement (than achievement) among young women after controlling for the socio-economic variables (especially education and employment) can be attributed to their incomplete fertility as they at least have 20 years to complete their reproductive span.

Some other findings of this study (not shown here) indicate that there is an inverse relationship between women's age and their age at marriage, therefore, it is not surprising to find a younger age at marriage to have an enhancing effect on the likelihood of overachieved fertility in comparison to achieved fertility. Table 8 shows that women who got married the youngest (under age 19) are 4.5 times more likely than women who got married at ages older than 25 to have the overachieved fertility in comparison to achieved fertility.

The multivariate analysis does not show any significant relationship between cousin marriage and the likelihood of achieving any categories of fertility gap. This finding is not

surprising as the bivariate tables showed that the correlation between women's marriage type (relative or non-relative) and fertility gap is weak. After controlling for demographic and socio-economic variables, the effect of cousin marriage on the likelihood of having over-achieved fertility disappears (Table 8).

After controlling for age, age at first marriage, cousin marriage, women's educational attainment and employment, women who have a balanced number of children are 90 percent less likely to have overachieved fertility, in comparison to achieved fertility than women who have more girls than boys. The bivariate analysis in Table 3 this study showed that there is a tendency among women to achieve a balanced sex composition of children (to have at least one child from each sex). As discussed, earlier women with more boys want to have more girls in the future. Conversely, women who have more girls wish to have boy in the rest of their reproductive span. The findings of Table 8 confirm this tendency even after controlling for the effect of other socio-economic variables still exist. The higher odds ratios of overachievement among women with more girls than women with a balanced sex composition implies that women with more girls than boys are more likely to continue to have children to reach a balanced sex composition among their children, and consequently are more likely to overachieve their fertility desire.

The multinomial logistic regression analysis goes on to show that after controlling for demographic and socio-economic variables, women who have a diploma and high school degree are 63 percent less likely than women with a bachelor's degree to have an overachieved fertility (in comparison to achieved fertility). As was shown in Table 5 women with high school and diploma degree and women with a university degree show the same trend in the achievement of the fertility gap. However, after considering the effect of other variables, the middle-educated women, in reference to women with university degrees, show a lower likelihood of overachievement (in comparison to achievement). It can, therefore, be concluded that due to some other social and especially economic factors (not considered in the logistic regression table in this study) women with high school and diploma restrict their fertility (more than women with university degrees).

Table 8 further shows that the likelihood of having overachieved fertility, in comparison to achieved fertility, is 60 percent less likely among employed women than unemployed women. The findings of Table 6 earlier too showed that there is a large difference in the proportion of overachievement between employed and unemployed women. Consistently, even after controlling for the interaction of demographic and socio-economic variables, employment has a depressing effect on the likelihood of overachievement (in comparison to achievement).

## Conclusion

Although there have been attempts to change Iran's national population policy from antinatalist back to pronatalist due to growing concerns regarding the negative consequences of long term below replacement fertility, the findings of this study in Tehran city, which has one of the lowest total fertility rates (TFRs) in Iran, reveal that if women can realise their intended number of children in addition of the number of children they currently have, they will be able to meet their fertility desires which is two children and more in this study. However, in case they face any socio-economic constraint, they would restrict their family size and finally have fewer children than they initially desired which might even be below replacement level. This paper also argues that despite the fact sex preference is disappearing even in more traditional regions of Iran, women who have more girls than boys are more likely to continue their childbearing to have a male child and eventually will have more children than they initially desired.

It is worth noting that in this study because of the time and budget constraints husbands were not included. However, men's role on fertility and their attitudes towards family planning

is very important. Therefore, for future studies investigating the family formation and reproductive behaviour it is recommended that men be involved in the sample selected for investigation.

## Author Contributions

**Conceptualization:** Maryam Hosseini, Udoy Saikia, Gouranga Dasvarma.

**Data curation:** Maryam Hosseini.

**Formal analysis:** Maryam Hosseini.

**Funding acquisition:** Maryam Hosseini.

**Investigation:** Maryam Hosseini, Udoy Saikia.

**Methodology:** Maryam Hosseini, Udoy Saikia, Gouranga Dasvarma.

**Project administration:** Maryam Hosseini, Udoy Saikia.

**Resources:** Maryam Hosseini.

**Supervision:** Udoy Saikia, Gouranga Dasvarma.

**Validation:** Maryam Hosseini.

**Writing – original draft:** Maryam Hosseini.

**Writing – review & editing:** Udoy Saikia, Gouranga Dasvarma.

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
