## [Decision Letter · Decision Letter 0]

4 Jan 2021

PONE-D-20-30495

The gap between desired and expected fertility among women in Iran: A case study of Tehran city

PLOS ONE

Dear Dr. Saikia,

Thank you for submitting your manuscript to PLOS ONE. After careful consideration, we feel that it has merit but does not fully meet PLOS ONE’s publication criteria as it currently stands. Therefore, we invite you to submit a revised version of the manuscript that addresses the points raised during the review process.

Considering the reviewers opinion, I am going with a decision of major revision for your paper. Along with a careful revision of the paper according to reviewer comments, the paper must be get it edited by a professional language editor before resubmission.

We look forward to receiving your revised manuscript.

Kind regards,

Srinivas Goli, Ph.D.

Academic Editor

PLOS ONE

Journal Requirements:

2.We note that you have indicated that data from this study are available upon request. PLOS only allows data to be available upon request if there are legal or ethical restrictions on sharing data publicly. For more information on unacceptable data access restrictions, please see http://journals.plos.org/plosone/s/data-availability#loc-unacceptable-data-access-restrictions.

Additional Editor Comments (if provided):

Considering the reviewers opinion, I am going with a decision of major revision for your paper. Along with a careful revision of the paper according to reviewer comments, the paper must be get it edited by a professional language editor before resubmission.

Reviewers' comments:

Reviewer's Responses to Questions

**Comments to the Author**

1. Is the manuscript technically sound, and do the data support the conclusions?

Reviewer #1: Yes

Reviewer #2: Partly

2. Has the statistical analysis been performed appropriately and rigorously? 

Reviewer #1: Yes

Reviewer #2: Yes

3. Have the authors made all data underlying the findings in their manuscript fully available?

Reviewer #1: Yes

Reviewer #2: Yes

4. Is the manuscript presented in an intelligible fashion and written in standard English?

Reviewer #1: Yes

Reviewer #2: Yes

5. Review Comments to the Author

Reviewer #1: This paper contains an interesting exercise on the achievement of desired and achieved fertility drawing the evidence from Iran’s capital, Tehran. Although this exercise needs longitudinal data, this paper tries to bring evidence from a cross-sectional survey. I have several observations on the current form of the paper which are the following.

1. Although the authors clearly mentioned how the ‘expected family size’, there is ambiguity on the ideal/desired number of children in the paper. How is the question of ideal number of children or desired family size asked during the survey? Does it follow the question on the Demographic and Health Survey (DHS) questionnaire? I assume that the question in the primary survey on the ideal number of children follows the DHS. In this kind of question, the post-factum rationalization is not accounted. For instance, a woman may say the ideal number of children is two in a low fertility setting immediately after marriage. If the woman already delivered three children at the age of 39 years (say after 20 years of marriage if the woman gets married at the age of 19 years), the woman may likely report the ideal number of children three. A long marital experience may affect the reporting of the number of the ideal number of children. This debate is missing from the paper.

2. The household decision-making may have a significant effect on the actual number of children. In-laws' family members such as the mother in law, father in law, and husband may be involved in deciding the number of children. In particular, husband’s involvement and women’s autonomy in reproductive decisions cannot be denied in any setting. Thus, the personal ideal number of childbearing and the actual number of childbearing (due to in-law’s pressure) may be different. This paper lacks this debate in the analysis and literature review.

3. Parity is an important aspect to understand the mismatch between desired family size and actual family size. Without the exercise of achievement of desired family size remain incomplete. Authors may include this variable in their analyses.

4. In a setting of low fertility and the existence of son preference (like Tehran), the majority of the couple may prefer one son and one daughter. In many cases, the couple may want to have at least one son. So, the categorization of the sample into three groups viz. the equal number of boys and girls, more boys than girls, and more girls than boys may not capture the true sense of son preference. Thus, a separate exercise categorizing the son preference variable into two groups such as have at least one son and don’t have a son may give an appropriate result.

5. A number of studies have documented that the excess of the actual number of children than the desired/ideal number of children is due to the lack of family planning services (both supply and demand-side). What do authors’ think about it? This part is missing the analyses as well as a discussion?

6. The results from multivariate analyses (the author did this analysis) are more reliable than the simple chi2 test. However, the authors comprehensively discussed the results from the chi2 test. The inferences from the chi2 test and multivariate analysis are similar but not the same.

7. Table 1: Since denominators are the same in the one age group (for instance 20-29), the summation of ‘mean CEB’ and ‘mean additionally the number of children’ should be the same as the ‘mean expected CEB’. Why is this inconsistency? This inconsistency is observed in some of the other tables too.

8. Please check the age categorization in table 2. (0-24 or 19-24).

9. In some places in the manuscript, double word (like Table 6 Table 6) and error message (Error! Reference source not found) appeared.

Reviewer #2: Thanks for giving me opportunity to read this paper “The gap between desired and expected fertility among women in Iran: A case study of Tehran city”. The research idea in not new but is relevant for policy and practice. Though the topic covered is impressive, the manuscript, however, falls short in terms of argument building, grammar, and notably in the interpretation section. The manuscript also suffers from empirical issues and hence must be rejected. The important observations of the paper are as follows,

1. In the research gap and setting, authors have failed to justify the need for conducting this study in Tehran city. What made authors to conducted such an important study in Tehran must be discussed thoroughly.

2. Author has mentioned that the study is based on the primary survey conducted on 400 currently married women. But how the author has reached to this sample size and what criterion were adopted to make samples representative of Tehran city needs proper explanation. A detail discussion on the survey design would be useful for readers.

3. It is quite surprising to notice that the Table 1 reports a sample of 401 married women aged 20-49 years from a total of 400 surveyed married women of age 15-49 years that to when 1 percent women aged 15-19 years were dropped from the study. This makes entire study dubious.

4. Again, in Table 3, heading in the first column is incorrect.

5. Some inferences drawn seem to be blanket statements, which need to be more nuanced. It is also preferable that the findings are discussed in the light of existing literature, especially if the results corroborate with what has been established earlier, for example, Women who have had delayed marriages have significantly higher levels of … fertility and consequently prevent overachieved fertility. Similarly, “The effect of age on women’s sex composition of children in this study shows that there is … sex composition of surviving children of women”. Similarly, at many place inferences are drawn from the analysis which is not the part of the study but from some other place of which this study is just a subset. For example, ‘The findings of this study show that most of the women …their preference for the sex of the next child (45 percent said it does not matter and 21 percent said it is up to God). This can create a huge confusion among the readers.

6. The present study may be suffering from post-facto rationalization based on which fertility at a given point in time is reported. An inclusion of limitation section would be better for the study.

7. An important finding highlighted from the study is that with increase in level of women’s education underachievement of the desired fertility is also increasing. Similarly, as the opinion on the gender of children becoming neutral higher share of women report highest underachievement in the desired fertility. Why is that ? Since this study is based on primary survey, it would be interesting if authors can provide reasons for such changes in the behaviour.

8. A descriptive table is needed to support the validity of the analysis conducted in the study.

It is recommended that authors consider substantial reworking on the reorganisation of the manuscript and resubmit it for further evaluation and necessary actions.

Best Wishes

6. PLOS authors have the option to publish the peer review history of their article (what does this mean?). If published, this will include your full peer review and any attached files.

Reviewer #1: **Yes: **Md Juel Rana

Reviewer #2: No

---

## [Author Response · Author response to Decision Letter 0]

23 Feb 2021

Reviewer #1: This paper contains an interesting exercise on the achievement of desired and achieved fertility drawing the evidence from Iran’s capital, Tehran. Although this exercise needs longitudinal data, this paper tries to bring evidence from a cross-sectional survey. I have several observations on the current form of the paper which are the following.

1. Although the authors clearly mentioned how the ‘expected family size’, there is ambiguity on the ideal/desired number of children in the paper. How is the question of ideal number of children or desired family size asked during the survey? Does it follow the question on the Demographic and Health Survey (DHS) questionnaire? I assume that the question in the primary survey on the ideal number of children follows the DHS. In this kind of question, the post-factum rationalization is not accounted. For instance, a woman may say the ideal number of children is two in a low fertility setting immediately after marriage. If the woman already delivered three children at the age of 39 years (say after 20 years of marriage if the woman gets married at the age of 19 years), the woman may likely report the ideal number of children three. A long marital experience may affect the reporting of the number of the ideal number of children. This debate is missing from the paper.

Author’s Response: 

The concepts of ideal and desired family size were introduced for general discussion, but data were collected only about desired family size based on the following question put to the women in the sample: 

“For you personally, what would be the number of children you would like to have in absence of any possible obstacle (economic, health…)?”. 

The women were asked a further question about their intended number of children, which formed the basis of calculating “expected fertility”. The question is as follows:

“In addition to the number of children you already have, how many (more) children you have intended to have in the rest of your childbearing period?”

However, any ambiguity about desired and ideal family size has been clarified in the text at the appropriate place by citing the definitions of ideal and desired family size. 

Discussions about issue of post-facto rationalisation raised by both the reviewers have now been added in detail in the highlighted text at the appropriate place (page 9 and 10).

2. The household decision-making may have a significant effect on the actual number of children. In-laws' family members such as the mother in law, father in law, and husband may be involved in deciding the number of children. In particular, husband’s involvement and women’s autonomy in reproductive decisions cannot be denied in any setting. Thus, the personal ideal number of childbearing and the actual number of childbearing (due to in-law’s pressure) may be different. This paper lacks this debate in the analysis and literature review.

Author’s Response: 

We greatly respect your comment, however it is worth noting that in the context of Tehran city which is a modern city with very high proportion of educated population, especially high proportion of educated women, the influence of in-law’s family member in reproductive decision making is weak. The concept of “household” in a city set up is very much dominated by nuclear family with less influence on decision making from other members of the family In fact, even previous studies in other capital cities show no sign of family member’s pressure on childbearing. An explanation to this effect has been added in the text (p.18)

3. Parity is an important aspect to understand the mismatch between desired family size and actual family size. Without the exercise of achievement of desired family size remain incomplete. Authors may include this variable in their analyses.

Author’s Response: 

We appreciate the reviewer’s comment and suggestion. We may point out that parity is taken into consideration in calculating expected fertility, because the specific question in this regard is:

“In addition to the number of children you already have, how many (more) children you have intended to have in the rest of your childbearing period?”, which does mention the number of children they already have (i.e. parity).

4. In a setting of low fertility and the existence of son preference (like Tehran), the majority of the couple may prefer one son and one daughter. In many cases, the couple may want to have at least one son. So, the categorization of the sample into three groups viz. the equal number of boys and girls, more boys than girls, and more girls than boys may not capture the true sense of son preference. Thus, a separate exercise categorizing the son preference variable into two groups such as have at least one son and don’t have a son may give an appropriate result.

Author’s Response: 

We truly appreciate and respect your comment. It seems a discussion is missing in the text that now I have added it (Page 15). The fact is that while a preference for son is rooted in Iranian culture, gender preference was not a driving force in the continuation of fertility in the context of low fertility of Tehran. I have added references to prove this argument.

5. A number of studies have documented that the excess of the actual number of children than the desired/ideal number of children is due to the lack of family planning services (both supply and demand-side). What do authors’ think about it? This part is missing the analyses as well as a discussion.

Author’s Response: 

Reviewer’s observation cannot be generalised for different country’s set up. In page 5 (Fertility transition and the emergence of below replacement fertility in Iran) it’s been explained that Iran experienced the most successful family planning program among developing countries. Therefore, having more children than desired cannot be attributed to lack of access to family planning program. We added a supporting discussion in page 26 (the third paragraph).

6. The results from multivariate analyses (the author did this analysis) are more reliable than the simple chi2 test. However, the authors comprehensively discussed the results from the chi2 test. The inferences from the chi2 test and multivariate analysis are similar but not the same.

Author’s Response:

We appreciate this comment. However, we would like to mention in this context that, while the Chi-Square tests were performed with five independent variables, each taken one at a time to examine their association with the dependent variable (and hence the length of discussion of the bivariate analysis). The statistically significant variables from the Chi-square analysis were then selected for multivariate analysis, which was performed with all the selected independent variables taken together. This accounts for the shorter length of discussion of the multivariate analysis, but it has, in no way left out the important points of the analysis.

7. Table 1: Since denominators are the same in the one age group (for instance 20-29), the summation of ‘mean CEB’ and ‘mean additionally the number of children’ should be the same as the ‘mean expected CEB’. Why is this inconsistency? This inconsistency is observed in some of the other tables too.

Author’s Response: 

The gap is calculated at individual level by subtracting the desired CEB from the expected CEB for each respondent. These are then averaged and presented in Table 1. This note is added below each 

8. Please check the age categorization in table 2. (0-24 or 19-24).

Author’s Response: 

The correct category is 19-24. The correction has been made.

9. In some places in the manuscript, double word (like Table 6 Table 6) and error message (Error! Reference source not found) appeared.

Author’s Response: 

Necessary corrections have been made.

 These errors are only shown in PDF file, however it’s Ok in the world file. Not sure where the error comes from? We can check it at before making the final PDF.

Reviewer #2: 

Thanks for giving me opportunity to read this paper “The gap between desired and expected fertility among women in Iran: A case study of Tehran city”. The research idea in not new but is relevant for policy and practice. Though the topic covered is impressive, the manuscript, however, falls short in terms of argument building, grammar, and notably in the interpretation section. The manuscript also suffers from empirical issues and hence must be rejected. The important observations of the paper are as follows. 

Authors’ response: 

Thanks for pointing this out. We have done a thorough document check for spelling and grammar. 

1. In the research gap and setting, authors have failed to justify the need for conducting this study in Tehran city. What made authors to conducted such an important study in Tehran must be discussed thoroughly. 

Author’s Response: On page (second paragraph) there is the justification about choosing Tehran city. A paragraph in track changes is also added.

2. Author has mentioned that the study is based on the primary survey conducted on 400 currently married women. But how the author has reached to this sample size and what criterion were adopted to make samples representative of Tehran city needs proper explanation. A detail discussion on the survey design would be useful for readers. 

Author’s Response: 

Thanks for your comment. An explanation, of how the sample size was selected is added on pages 10 and 11.

3. It is quite surprising to notice that the Table 1 reports a sample of 401 married women aged 20-49 years from a total of 400 surveyed married women of age 15-49 years that to when 1 percent women aged 15-19 years were dropped from the study. This makes entire study dubious. 

Author’s Response: 

The age distribution in groups of the interviewed women are given in Table 1. As clearly it was stated that married women in 15-19 age group is extremely small and therefore negligible (only one percent), statistically it does not add any value to include this as a group, and hence this group was dropped from analysis which is clearly stated in paragraph 1 of page 12. 

4. Again, in Table 3, heading in the first column is incorrect. 

Author’s Response: 

Thanks. Correction has been made accordingly. 

1. Some inferences drawn seem to be blanket statements, which need to be more nuanced. It is also preferable that the findings are discussed in the light of existing literature, especially if the results corroborate with what has been established earlier, for example, Women who have had delayed marriages have significantly higher levels of … fertility and consequently prevent overachieved fertility. Similarly, “The effect of age on women’s sex composition of children in this study shows that there is … sex composition of surviving children of women”.

 Similarly, at many place inferences are drawn from the analysis which is not the part of the study but from some other place of which this study is just a subset. For example, ‘The findings of this study show that most of the women …their preference for the sex of the next child (45 percent said it does not matter and 21 percent said it is up to God). This can create a huge confusion among the readers. 

Authors’ response: 

(1) Re the comment: “Women who have had delayed marriages have significantly higher levels of … fertility and consequently prevent overachieved fertility”. We can only say in response that the statement quoted by the reviewer simply interprets the finding why women with delayed marriages have been able to prevent over-achieving their fertility desire.

(2) Re the comment: “The effect of age on women’s sex composition of children in this study shows that there is … sex composition of surviving children of women”. Again, we have interpreted the finding from our analysis. However, we have modified the sentence in question on p. 21.

(3) Re the comment: “The findings of this study show that most of the women …their preference for the sex of the next child (45 percent said it does not matter and 21 percent said it is up to God).” We have removed references to 45 percent and 21 percent (p.19), which we hope will not create any confusion.

6. The present study may be suffering from post-facto rationalization based on which fertility at a given point in time is reported. An inclusion of limitation section would be better for the study. 

Authors’ response: 

We have provided a response to a similar comment made by the first reviewer. 

1. An important finding highlighted from the study is that with increase in level of women’s education underachievement of the desired fertility is also increasing. Similarly, as the opinion on the gender of children becoming neutral, higher share of women report highest underachievement in the desired fertility. Why is that? Since this study is based on primary survey, it would be interesting if authors can provide reasons for such changes in the behaviour. 

Author’s Response: The comment is applied in the first paragraph of page 19 and page 21.

A descriptive table is needed to support the validity of the analysis conducted in the study. 

Author’s Response: The following table now has been added in page no 13. 

Women's mean age 34.5

Women’s mean age at first marriage 22.8

The mean CEB of women 1.38

Women's level of education

Elementary 21.9

High-Diploma 32.9

Bachelor's degree and above 45.1

Employment status of women 

Employed 45.2

Unemployed 54.8

---

## [Decision Letter · Decision Letter 1]

28 Apr 2021

PONE-D-20-30495R1

The gap between desired and expected fertility among women in Iran: A case study of Tehran city

PLOS ONE

Dear Dr. Saikia,

Thank you for submitting your manuscript to PLOS ONE. After careful consideration, we feel that it has merit but does not fully meet PLOS ONE’s publication criteria as it currently stands. Therefore, we invite you to submit a revised version of the manuscript that addresses the points raised during the review process.

ACADEMIC EDITOR: Reviewers are in favour of recommending this piece. However, reviewers suggest some minor revisions to the manuscript. I recommend authors to consider those minor suggestions. 

We look forward to receiving your revised manuscript.

Kind regards,

Srinivas Goli, Ph.D.

Academic Editor

PLOS ONE

Journal Requirements:

Additional Editor Comments (if provided):

Reviewers are in favour of recommending this piece. However, reviewers suggest some minor revisions to the manuscript. I recommend authors to consider those minor suggestions.

Reviewers' comments:

Reviewer's Responses to Questions

**Comments to the Author**

1. If the authors have adequately addressed your comments raised in a previous round of review and you feel that this manuscript is now acceptable for publication, you may indicate that here to bypass the “Comments to the Author” section, enter your conflict of interest statement in the “Confidential to Editor” section, and submit your "Accept" recommendation.

Reviewer #1: (No Response)

2. Is the manuscript technically sound, and do the data support the conclusions?

Reviewer #1: Yes

3. Has the statistical analysis been performed appropriately and rigorously? 

Reviewer #1: Yes

4. Have the authors made all data underlying the findings in their manuscript fully available?

Reviewer #1: Yes

5. Is the manuscript presented in an intelligible fashion and written in standard English?

Reviewer #1: Yes

6. Review Comments to the Author

Reviewer #1: Although authors have tried to address all comments, two comments have not been appropriately addressed. Against the comment 3 and 4, authors did not give suitable responses and did not adequately make necessary changes in the manuscript. Authors are advised to analyse two things and include them in the paper. The level of mismatch of desired fertility and actual fertility by parity and sex-composition (using an additional definition). Thus, the authors may estimate the mean CEB, mean additionally intended number of children, mean expected CEB, mean desired family size, and gaps by the number of children delivered (parity) such as 1, 2, and 3+. Similarly, the same estimates may be produced by the women who have at least one son and who don’t have a son.

I would like to review this manuscript again with the results of the above-mentioned analyses.

7. PLOS authors have the option to publish the peer review history of their article (what does this mean?). If published, this will include your full peer review and any attached files.

Reviewer #1: **Yes: **Md Juel Rana

---

## [Author Response · Author response to Decision Letter 1]

29 Jun 2021

Reviewer #1: Although authors have tried to address all comments, two comments have not been appropriately addressed. Against the comment 3 and 4, authors did not give suitable responses and did not adequately make necessary changes in the manuscript. Authors are advised to analyse two things and include them in the paper. The level of mismatch of desired fertility and actual fertility by parity and sex-composition (using an additional definition). Thus, the authors may estimate the mean CEB, mean additionally intended number of children, mean expected CEB, mean desired family size, and gaps by the number of children delivered (parity) such as 1, 2, and 3+. Similarly, the same estimates may be produced by the women who have at least one son and who don’t have a son.

Author’s response: The analysis suggested by Reviewer #1 has been added in the form of. two tables - Table 4 and Table 7 and their interpretation, to address the level of mismatch of desired fertility and actual fertility by women’s parity and sex-composition of their children. There was no further comment from Reviewer#2.

---

## [Editor Report · Decision Letter 2]

2 Jul 2021

PONE-D-20-30495R2

The gap between desired and expected fertility among women in Iran: A case study of Tehran city

PLOS ONE

Dear Dr. Saikia,

Thank you for submitting your manuscript to PLOS ONE. After careful consideration, we feel that it has merit but does not fully meet PLOS ONE’s publication criteria as it currently stands. Therefore, we invite you to submit a revised version of the manuscript that addresses the points raised during the review process.

ACADEMIC EDITOR: The paper can be considered for publication in PLOS One. However, in its current formatting (e.g. Abstract and referencing style) do not follow PLOS author guidelines. Thus, sending back to authors. Please format the paper according to PLOS One author guidelines and resubmit it. 

We look forward to receiving your revised manuscript.

Kind regards,

Srinivas Goli, Ph.D.

Academic Editor

PLOS ONE

Journal Requirements:

Additional Editor Comments (if provided):

The paper can be considered for publication in PLOS One. However, its current formatting (e.g. Abstract and referencing style) do not follow PLOS author guidelines. Thus, sending back to authors. Please format the paper according to PLOS One author guidelines and resubmit it.

---

## [Author Response · Author response to Decision Letter 2]

26 Jul 2021

We have already responded to the reviewers comments and submitted the manuscript with changes. This came back with some minor changes required in formatting as mentioned in comments section above.

---

## [Editor Report · Decision Letter 3]

25 Aug 2021

The gap between desired and expected fertility among women in Iran: A case study of Tehran city

PONE-D-20-30495R3

Dear Dr. Saikia,

We’re pleased to inform you that your manuscript has been judged scientifically suitable for publication and will be formally accepted for publication once it meets all outstanding technical requirements.

Kind regards,

Srinivas Goli, Ph.D.

Academic Editor

PLOS ONE
---

## [Editor Report · Acceptance letter]

6 Sep 2021

PONE-D-20-30495R3 

The gap between desired and expected fertility among women in Iran: A case study of Tehran city 

Dear Dr. Saikia:

I'm pleased to inform you that your manuscript has been deemed suitable for publication in PLOS ONE. Congratulations! Your manuscript is now with our production department. 

Kind regards, 

on behalf of

Dr. Srinivas Goli 

Academic Editor

PLOS ONE